# Investigating the Disordered and Membrane-Active Peptide A-Cage-C Using Conformational Ensembles

**DOI:** 10.3390/molecules26123607

**Published:** 2021-06-12

**Authors:** Olena Dobrovolska, Øyvind Strømland, Ørjan Sele Handegård, Martin Jakubec, Morten L. Govasli, Åge Aleksander Skjevik, Nils Åge Frøystein, Knut Teigen, Øyvind Halskau

**Affiliations:** 1Department of Biological Sciences, University of Bergen, N-5020 Bergen, Norway; odo021@uib.no (O.D.); oyvind.stromland@uib.no (Ø.S.); orjan.handegard@gmail.com (Ø.S.H.); martin.jakubec@uit.no (M.J.); m.larsen@ucl.ac.uk (M.L.G.); 2Department of Biomedicine, University of Bergen, N-5020 Bergen, Norway; age.a.skjevik@gmail.com (Å.A.S.); knut.teigen@uib.no (K.T.); 3Division of Infection and Immunity, University College London, London WC1E 6BT, UK; 4Department of Chemistry, University of Bergen, N-5020 Bergen, Norway; nils.froystein@uib.no

**Keywords:** oligomerizing peptides, bicelles, lipids, conformation, slow exchange, ensemble clustering, misfolding

## Abstract

The driving forces and conformational pathways leading to amphitropic protein-membrane binding and in some cases also to protein misfolding and aggregation is the subject of intensive research. In this study, a chimeric polypeptide, A-Cage-C, derived from α-Lactalbumin is investigated with the aim of elucidating conformational changes promoting interaction with bilayers. From previous studies, it is known that A-Cage-C causes membrane leakages associated with the sporadic formation of amorphous aggregates on solid-supported bilayers. Here we express and purify double-labelled A-Cage-C and prepare partially deuterated bicelles as a membrane mimicking system. We investigate A-Cage-C in the presence and absence of these bicelles at non-binding (pH 7.0) and binding (pH 4.5) conditions. Using in silico analyses, NMR, conformational clustering, and Molecular Dynamics, we provide tentative insights into the conformations of bound and unbound A-Cage-C. The conformation of each state is dynamic and samples a large amount of overlapping conformational space. We identify one of the clusters as likely representing the binding conformation and conclude tentatively that the unfolding around the central W23 segment and its reorientation may be necessary for full intercalation at binding conditions (pH 4.5). We also see evidence for an overall elongation of A-Cage-C in the presence of model bilayers.

## 1. Introduction

Protein amphitropism is a phenomenon where soluble proteins, usually globular, reversibly interact with the cellular plasma membrane. The reversibility is caused by ligand binding, post-translational modification [1], or local changes in the environment, such as changes in temperature or pH [2,3,4]. Amphitropic proteins are essential for many cellular functions, such as cytoskeletal reorganization, the lateral organization of the cell membrane, and its biogenesis, remodelling and maintenance [2,5,6,7,8]. Amphitropism is also involved in numerous enzymatic processes [3,9,10] and membrane-associated cell signalling and transduction [7,9,10,11]. Moreover, amphitropic proteins such as tyrosine hydroxylase, monoamine oxidase A, and neutrophil serine proteases are important in biomedical and pharmacological research [12,13,14]. To switch between a solubilized and a membrane-associated state, amphitropic proteins must be prone to reorganization. Usually, intrinsic polypeptide disorder and weakly folded domains are common features of amphitropic proteins [4,15]. At the level of protein secondary structure, the presence of flexible amphipathic helices in a variable protein environment, accompanied by fine-tuning of their charge, hydrogen bonds and π-cation interactions are important for protein–membrane interactions [16,17,18,19]. Such features have been found in many amphitropic proteins including α-Lactalbumin (ALA), EBP1, lipoxygenases, 14-3-3γ, Tyrosine Hydroxylase and α-Synuclein [4,7,20,21,22].

Intrinsic disorder is also overrepresented in proteins that form in vivo aggregates of different types, including amorphous aggregates and b-sheet-rich amyloid fibrils [23]. The disease-associated amyloid aggregates often contain lipids [24,25], and the amyloid misfolding toxicity mechanisms are closely associated with protein–membrane interaction [26,27]. Investigation of model systems for protein–membrane interactions are therefore of interest. ALA, a substrate-specifier of galactosyltransferase expressed during lactation in mammals [28], interacts reversibly with acidic lipid bilayers in a charge-dependent manner [29,30]. This interaction occurs primarily through two helices referred to as Helix A and Helix C [18,31]. These helices showed increased protection from solvent exchange compared to native and molten-globule controls when the protein was bound to negatively charged vesicle membranes (Figure 1A), indicating that these were key structural elements in the lipid–protein interactions [18,32]. Helix A and C, either as individual helical peptides or linked together, present interesting interfacial properties, including disruption of the bilayer through either a membrane-thinning or a pore-like mechanism associated with protein aggregation [33,34].

Here, we investigate the chimeric A-Cage-C polypeptide, one of the artificially designed recombinant peptides previously investigated by Strømland et al. [34]. A-Cage-C joins the A and C helices of bovine α-Lactalbumin (BLA) using the Tryptophan Cage, a small folding unit (Figure 1B) [35]. Our motivation for studying it was initially as a model system for membrane-induced cytotoxic protein oligomerization. However, while the polypeptide did form aggregates in the presence of solid-supported bilayers [34], the link to amyloid action is uncertain, and the peptides did not exhibit any significant toxicity. The peptide did show a pH-dependent membrane binding similar to its parent protein (Figure 1) [18,29,34], and as such, it is a model system for pH-dependent amphitropism. In principle, it could also find use as a membrane-targeting drug carrier in a manner similar to its parent protein [36]. Our motivation for performing this study is to shed light on molecular mechanisms for membrane interaction and aggregation mechanisms. Our investigation starts with a simple in silico analysis of A-Cage-C. We then attempt to chart the overlap between the fluctuating structure in solution and association with disc-like bicelles, a membrane-mimicking system, at neutral and acidic pHs, using ^31^P NMR, triple-resonance NMR, conformational clustering, and molecular dynamics (MD). We comment critically on the suitability of the model system and our approaches.

## 2. Results

### 2.1. Selected Predicted Properties of A-Cage-C

To initiate the study, we performed in silico assessments of the A-Cage-C sequence. We calculated its hydropathy index and secondary structure propensity using the approach of Tanford and Protparam, respectively [37,38]. Then we predicted three indices of polypeptide disorder using DisEMBL [39]. Finally, we used AMYLPRED2, a computational tool that integrates up to 11 methods to identify amyloidogenic segments within an input amino-acid sequence [40]. Overall, A-Cage-C is predicted to be only somewhat disordered when considering the HOTLOOPS and REMARK456 indices. These metrics compare the query sequence to sequences that tend to have a B-factor or missing electron densities in the PDB databank (Figure 2A). There are intervals of intermediate to high hydropathy (L14-l24 and P34-V50), suggesting that these segments would intercalate into a hydrophobic lipid environment if present. Such hydrophobic regions are also implicated in protein misfolding and aggregation [23,41], where they serve as nucleation sites for intermolecular connections. The region between these intervals, centred around S30 and corresponding to the Tryptophan-Cage part of the polypeptide, exhibits both negative hydropathy and propensity for the disorder. Within the A-Cage-C sequence, Helix A and Helix C are predicted by PSIPRED [38] to be extended and shortened, respectively, as indicated in the Figure 2B. AMYLPRED2 predicts two polypeptide chain segments to be prone to amyloid formation; L17-L24 and M48-A49 (Figure 2B). These lie with the predicted helices and intervals of high hydropathy.

### 2.2. Isotope Labelling of A-Cage-C and Preparation of Bicelles

NMR is one of the most powerful methods available to resolve individual atoms and their chemical environment in solution [42]. However, obtaining high-resolution NMR on polypeptides interacting with proton-rich, large lipid membrane systems is challenging due to enhanced dipole–dipole relaxation, exchange broadening, and signal overlap [43,44]. To experimentally explore the conformational propensity of this polypeptide in the presence and absence of a membrane-mimicking system, two components needed to be acquired—^13^C and ^15^N labelled A-Cage-C, and a suitable membrane mimicking system with a low proton background. To that end, double labelled A-Cage-C was expressed and purified by adapting a protocol published earlier [34], as described in Materials and Methods. After optimizing the expression system, we could purify labelled A-Cage-C from the soluble fraction of lysed cells (Figure 3A). The peptide’s affinity tag was cleaved off and purified further using size exclusion chromatography to produce pure peptide (Figure 3B).

We chose bicelles for the membrane mimicking part of the two-component system. These are similar to micelles, except that they have disc-like shapes consisting of a bilayer protected by a periphery of short, detergent-like phospholipids [45]. In bicelles, the lipids are in exchange with the solvent, and bicelle morphologies are dependent on the lipid mixing ratio, *q*, as well as solvent conditions and interacting polypeptides [46,47]. Despite this, we picked this system over non-exchanging nanodiscs as it proved unpractical to produce samples with high enough A-Cage-C concentration in the presence of excess amounts of nanodiscs. We also experienced sample precipitation when nanodiscs were added (data not shown) and abandoned the nanodisc approach. Small bicelles (with *q* values < 2) are isotropically fast-tumbling membrane-mimicking systems and are suitable for studying many aspects of membrane–protein interactions [48,49,50]. We, therefore, prepared bicelles with a *q* = 0.4 from deuterated D22-DHPC, deuterated D54-DMPC, DHPS and DMPS as described in Materials and Methods and [51,52], and assessed their morphology using Transmission Electron Microscopy (TEM) (Figure 4A). Although the TEM images were sparsely populated and bicelles are adversely affected by staining, dry conditions and salt from buffers, the micrographs did suggest the presence of disc-shaped particles (Figure 4A inset). Their mean diameter was 5.9 ± 0.7 nm, based on 35 distance measurements using the ImageJ software. We then performed ^31^P NMR on the bicelles in the presence and absence of A-Cage-C (Figure 4B), and assigned the resolved resonances. The most intense signal was assigned to the DHPC component which is present at highest abundance in the bicelles. DMPC, the second most abundant component, is the peak immediately upfield of this. The resonances of the minor components, DHPS and DMPS, appears to overlap, suggesting that their chemical environment is indistinguishable. The relative positioning of the PS and PC components are consistent with what is known from literature [53]. The presence of A-Cage-C (Figure 4B bottom traces) changes the integration values, and shifts and broadens the resonances somewhat, suggesting that a perturbing interaction takes place. Taken together, we conclude that bicelles were produced and A-Cage-C interacts with them at pH 4.5.

### 2.3. pH-Induced Conformational Changes

Earlier work showed that the A-Cage-C bound to acidic lipid membranes at pH 4.5 and that at pH 7.0 this binding was practically absent [34]. In addition to residue- and lipid headgroup protonation states [18,32], pH could also influence the polypeptide conformation and, therefore, its ability to interact, e.g., by making hydrophobic surfaces more accessible for interaction [18,29]. We compared the conformational propensities for the two different pHs at the residue level to investigate this possibility. We acquired ^1^H^15^N HSQC fingerprints and triple-resonance spectra in the presence and absence of bicelles at binding (pH 4.5) and non-binding (pH 7.0) conditions. Figure 5A,B show ^1^H-^15^N HSQC spectra of A-Cage-C acquired at pH 4.5 in the absence of bicelles and the presence of bicelles, respectively. Data acquired at pH 7.0 is presented in Appendix A. The HSQC signal distributions are in all cases narrow along the proton axis, and all spectra display broad signals (Figure 5 and Appendix A). This observation confirms that A-Cage-C does not have a well-defined fold, and is dominated by relatively slow conformational fluctuations and chemical exchange [43,44]. We first assigned the polypeptide resonances in the absence of bicelles at pH 7.0. The spectra obtained allowed us to assign 33 out of 52 non-proline residues (Appendix A). Then, for binding conditions at pH 4.5, resonance assignment revealed an additional set of minor signals predominantly in the C-terminus (E3, G32, A49, K51, K52, L54, D55, K56, G58) (Figure 5A). The appearance of such signal is the result of an exchange between conformational states with rates that are slow on the NMR time scale [44], and the presence of additional signals describing the same residue indicates that a second, and in some cases, even a third, conformation is available for these residues through a slow exchange. The assignment was then performed at binding conditions (pH 4.5) in the presence of bicelles (Figure 5B). In this case, resonances were broadened even further, many of them beyond detection (Figure 5B). We interpret this as an association of A-Cage-C with the bicelle, where additional broadening comes from an exchange between bound and unbound states [54,55]. 

There may also be efficient dipole–dipole relaxation from the remaining proton background of the partially deuterated bicelles [43]. The resonance assignments obtained for bound and unbound A-Cage-C situations were deposited in the Biological Magnetic Resonance Data Bank (BMRB) under the accession code 26892. Comparing the fingerprints and shifts of both pHs in the absence of bicelles showed perturbations at the N-terminal helix and C-terminal helix (Appendix A), suggesting that conformational change takes place in these regions that correspond to part of the A Helix and most of the C Helix. These motifs are also involved in the binding of the full-length protein [18,33,34].

Although the assignment was incomplete, the determined Cα and Cβ chemical shifts can be used to investigate the secondary structure propensity of A-Cage-C in its different states using the Secondary Structure Propensity (SSP) program [23]. The SSP analysis indicated that A-Cage-C is α-helical at S2-V10, Q22–27 G in the Cage motif, L39-D46. The most notable change as the polypeptide shifts from pH 7.0 to pH 4.5 is a weakening of helicity in the Cage motif (Figure 6A,B). In contrast, there are relatively minor changes in the two helices. The addition of bicelles, and the subsequent comparison of the resulting spectrum with the bicelle-free situation, showed both chemical shift changes and signal broadening beyond detection for all resonances detected in the bicelle-free case (Figure 5A,B). The secondary conformations of the residues indicated to be in slow exchange were also broadened (Figure 5A,B). Analysis by chemical shift perturbation indicated that Helix A and C showed perturbation near the N-terminal and C-terminal, respectively, and disappearance of signals due to signal broadening closer to the center of A-Cage-C (Figure 6C). The lack of identifiable signals precluded analysis of the central regions, but the D26-R33 segment, within the Cage region of A-Cage-C, experienced a shift indicative of binding. An SSP analysis of the bicelle-associated states at pH 4.5 was not performed due to a lack of Cα and Cβ resonance assignments in this state.

### 2.4. Structural Ensembles in the Presence and Absence of Bicelles

The incomplete sequential assignment and a paucity of interresidual Nuclear Overhauser Effects (NOEs) precluded traditional structural assessment using distance constraints. Therefore, an alternative approach using the calculation of structural ensembles was undertaken. Briefly, structural representations of A-Cage-C in both its free and bicelle-associated forms were produced based on SSP and TALOS N dihedral angle data. These were then incorporated into the Flexible-Meccano software [56]. For each experimental condition, 10,000 conformers were produced. These conformational ensembles were further divided into clusters using hierarchical agglomerative clustering. A number of clusters accounting for as much data as possible while maintaining a low number of clusters were picked using the Davies–Bouldin Index (DBI) [57,58] and pseudo-F-statistic (pSF) [58] metrics (See Appendix A and associated comment). The statistical evaluation suggested that two clusters accounted for nearly all conformations (>99.99%). Therefore, we proceeded with Cluster 1 and 2 based on data corresponding to the absence and presence of bicelles at binding condition (pH 4.5). 

The representative conformations of the two clusters (centroids) obtained for the free peptide are presented in Figure 7A. An α-helix appearing near the C-terminus is featured in both Cluster 1 and 2, both predominantly disordered (Figure 7A). The primary differences between the clusters are relative helix orientations, the appearance of a helical tendency near the central tryptophan W23 and a change in this residue orientation in Cluster 2. This is not entirely in line with the initial SSP data (Figure 6A,B), which showed α-helical propensity in several parts of A-Cage-C. Centroid representations of A-Cage-C’s two clusters in the bicelle-bound situation demonstrate also an α-helical propensity in the C-terminus, as well as the appearance of a new helix in the N-terminus (residues G1-F11) (Figure 7B). This is stronger for Custer 1, and we observed a minor helical conformation near W23 in this cluster, similar to Cluster 2 for the absence of bicelles. The orientation of this residue is also affected. It should be stated that the distributions of conformations around the centroids of all representations are broad. However, inspecting all conformations also suggests that Cluster 1 of both states are elongated relative to Cluster 2 (Appendix A).

### 2.5. Molecular Dynamics Simulations in the Presence of Bilayers

Based on incomplete experimental data, the cluster analysis provided us only with quite tentative models for states that may be relevant for A-Cage-C binding. In an attempt at complementing these models with an independent approach, we performed MD simulations of A-Cage-C in the presence of lipid bilayers. To find a representative A-Cage-C starting conformer, preparative simulations were performed on two A-Cage-C representations picked from an ensemble of 10,000 structures generated using Flexible-Meccano. The most compact and the most extended structures were treated in silico as described in Materials and Methods and simulated in explicit water for up to 520 ns (Appendix A). The DSSP secondary structure analysis of the simulations indicate that A-Cage-C starts out as dominated by loops and becomes increasingly helical in the C-terminal region of the polypeptide towards the latter half of the simulation time. The appearance of stable α-helical structures coincides only somewhat with the initial sequence analysis presented in Figure 2 (Appendix A) and with the centroid structures of the cluster analysis (Figure 7). In the simulation using the most compact form as a starting point, there is also evidence of extended structures in the Q22-S32 region (Appendix A). This secondary structure type is indicative of β-sheet propensity and lies close to a region indicated by AMYLPRED2 as prone to form amyloids (L19-L24, Figure 2B). Yet, there is no experimental evidence in this study indicating the formation of β-sheets. Where SSP data is available, A-Cage-C has a strong α-helical propensity in this region (Figure 6). 

The two simulations represent extremes within a structurally diverse ensemble, and because of this, they did not converge within the simulation time (Appendix A). Instead of extending the simulations indefinitely, we combined all frames of both simulations and clustered them using the same approach as described above. We then picked a starting structure from the largest cluster that had the lowest cumulative distance to all other frames within its cluster. This starting structure was used to set up two simulations: A-Cage-C in the presence of a DOPC bilayer and A-Cage-C in a DOPC:DOPS (4:1) bilayer. The simulations were run for 800 ns in each case, and root-means-square deviation (RMSD) plots vs simulation frame indicate that equilibrium is not reached (Figure 8A,B). Moreover, A-Cage-C does not settle at or within the bilayer but fluctuates between the bilayer interface and the aqueous bulk. These simulations were performed at neutral pHs using the TIP3P, as we lack proper parametrization of lipid headgroups for other conditions. Despite this, we allow ourselves to note that DOPC and DOPC:DOPS bilayers affect A-Cage-C differently. In the DOPC case, the DSSP secondary structure analysis is similar to that of the simulations in water only: A-Cage-C is dominated by coils and fluctuating helices (Figure 8C). It may be argued that the helical propensity is somewhat stronger when DOPC is present. Contrasting DOPC to the DOPC:DOPS simulation, we observe that the latter simulation has more helical conformations interspersed with persistent extended conformations towards the end of the simulation (Figure 8D).

## 3. Discussion

The previous study using CD and fluorescence support that A-Cage-C is somewhat helical in solution at both pHs. When bound to bilayers, this helicity increases, and there is evidence of coils but no β-sheets [34]. In silico predictions based on primary sequence, NMR-based SSP, Flexible-Meccano cluster-analysis, and MD simulations are only in partial agreement with respect to secondary structure assignments. In contrast to CD, fluorescence, MD and NMR, the in silico prediction suggests a higher helical content. However, the tendency towards disorder indicated by the primary sequence analysis is in line with the large, diffuse and overlapping conformational clusters derived from the cluster analysis, as well as the broadened and slowly exchanging NMR signals observed, especially at pH 4.5. Moreover, while the in silico analysis indicates that β-amyloid formation can occur, we find little evidence from any of the experimental techniques used here that A-Cage-C can sample β-sheet conformations, a likely requirement for initiating β-amyloid formation. The primary exception is the MD simulations at neutral pHs in the presence of DOPC:DOPS bilayers, where extended conformations or β-sheets do occur. However, our tentative models for the bicelle-associated state do not support this, although it should be stated that we lack experimental insight from SSP or TALOS N for significant parts of the A-Cage-C sequence.

Isotropically tumbling bicelles with relatively low *q*-values have been criticized for being fluctuating and not always retaining a disc-like morphology [46,59]. We operated with a *q*-value in the upper range of what could be considered small bicelles, and at least some evidence for disc-like morphologies could be found in the TEM micrographs (Figure 4A). Moreover, both the ^31^P and the triple-resonance NMR-data suggests A-Cage-C interaction. This is evidenced by a slight downfield shift of ^31^P resonances, changes in integral ratios of the DMPC/DMPS and DHPC signals, and line broadening of both ^31^P and ^1^H resonances (Figure 4B and Figure 5C). These observations are diagnostic of peptide interaction in similar bicelle-peptide interaction systems [60,61]. Overall, the evidence suggests that this interaction may be superficial and transient in nature. In the MD simulations, A-Cage-C does not settle at or within the bilayer but fluctuates between the bilayer interface and the aqueous bulk. This is likely due to electrostatic repulsion between A-Cage-C, and the lipids are not overcome. Interaction of ALA and A-Cage-C requires acidic conditions to protonate key residues in Helix A and Helix C [18,34], and it may be that our simulation is not able to capture these events sufficiently well. Also, solvation using the TIP3P water model may lead to excessively compact A-Cage-C structures, as has been reported seen in the cases of other disordered peptides [62,63]. Compact structures might hamper interaction by reducing bilayer access to key binding elements of A-Cage-C.

Although triple-resonance NMR data acquired on A-Cage-C proved difficult to assign and structurally analyse through standard methods, Flexible-Meccano and MD simulations provided us with some structural data. These methods also provided us with a coarse view of the very large conformational space available for the peptide to sample in different states. In our previous work, the central tryptophan of the Cage motif failed to intercalate into the membrane for interactions that do not cause membrane leakage, which was associated with perturbation of the bilayer core [34]. The unfolding of the central Cage motif may be a prerequisite for intercalating this segment, and conditions that favour this would promote such intercalation. In line with this, we see traces of structure similar to the Trp-Cage fold left in one of the centroid representations, i.e., a short helix and proline-rich loops loosely surrounding the central tryptophan [35]. In the MD simulations, the Trp-Cage fold may be present in simulation in water only (Appendix A), absent in the DOPC case (Figure 8C), and coinciding with an extended conformation in the DOPC:DOPS case (Figure 8D). A Cage-like structure is seen in Cluster 2 in the associated situation (Figure 7A), and Cluster 1 in the bicelle-associated situation (Figure 6B). However, the disappearance of the residual Trp-Cage fold does not coincide with the appearance of longer helices (Figure 7B), as would also be expected for the membrane-bound conformation. The simplest explanation is that the bound situation samples the overlapping conformational space around each centroid representation, and no additional helicity is seen explicitly in our approach. The disappearance and extreme broadening of resonances observed when bicelles are present are consistent with such behaviour. Another possibility is that the minor conformations apparent in the NMR data at pH 4.5 in the absence of bicelles may represent one of the clusters, i.e., Cluster 2. 

A-Cage-C showed only weak tendencies to form aggregates at lipid bilayers supported on a solid mica surface [34], and these aggregates did not show the fibrillar nature of amyloid aggregates. However, they did coincide with the ability of A-Cage-C to cause membrane leakage, a property associated with amyloid oligomers [42,60], both also with e.g., membrane thinning or helical bilayer disruption mechanisms [64,65]. While MD simulations suggest the ability of A-Cage-C to sample extended conformations in the presence of DOPC:DOPS bilayers (Figure 8D), we observed practically no β-sheet tendencies in our experimental results. The samples we used were stable within the NMR project timeline (weeks), suggesting that aggregation of any kind did not easily occur. ALA, from which A-Cage-Cs helical motifs are derived (Figure 1A), shows superficial binding for liquid-ordered membrane phases. At the same time, the helices have better access to the hydrophobic core of the membrane for fluid bilayers [32,66]. The bicelles may have been protective towards aggregation by shielding hydrophobic nucleation sites. Of possible relevance, such behaviour is seen for both zwitterionic and anionic fluid bilayers in the case of α-Synuclein [54,67], where the presence of such bilayers inhibits amyloid aggregation. In contrast, lipids organized in the liquid-order phase seem important for misfolding events leading to α-Synuclein oligomers [54,68].

MD simulations in the presence of DOPC:DOPS bilayers induce extended conformations (Figure 8D), and two clusters are distinguished by relative elongation (Appendix A). Such elongation, when taking place parallel to a bilayer, could be part of a lipid-induced misfolding mechanism [54,69]. However, beyond this, we see no fold tendencies in A-Cage-C that suggests misfolding and oligomerization. Our experimental data and MD simulations indicate that the interaction of A-Cage-C with model membranes is superficial and dynamic, which is expected for pH 7.0. At pH 4.5 we primarily rely on NMR data, but also here the data suggest superficial interaction. It may be that bicelles fail to provide and stabilize the binding mode necessary to promote inter-chain hydrophobic contacts and possibly β-sheet organization that could facilitate initiation of aggregation or amyloid formation [39,41]. Instead of inducing β-strands, the fold remains soluble and with an α-helical tendency in the presence of bicelles, and amyloid nucleation sites are protected by embedding into the membrane. Indeed, the lamellar lipid packing of bicelles has been reported to contain defects stemming from a minor component of the short-chain lipids primarily sequestered along the putative disc rim [59]. This may make it easier for a polypeptide to access the disc bilayer’s hydrophobic core, and in line with a greater soluble accessible surface area that has been reported for bicelles with *q*-values similar to those we use [46].

We tentatively conclude that Cluster 1 associated with the bicelle situation may loosely reflect the interacting conformation. In this, we observe a shift towards A-Cage-C elongation and induction of a long N-terminal helix. In support of this is the observation that α-helical induction is also observed in the last parts of the DOPC:DOPS simulations, which were affected most by the presence of a bilayer. This state does not unfold the residual Trp-Cage fold but depending on W23 placement relative to the membrane, it may still be able to intercalate. Despite in silico prediction and previous work suggesting that aggregation takes place, no evidence for bicelle-driven A-Cage-C aggregation were observed. This indicates that A-Cage-C is not a good system for exploring amyloid aggregation or that bicelles fail to induce appropriate starting conformations for amyloid formation.

## 4. Materials and Methods

Egg yolk phosphatidylcholine (EYPC, 99% purity) and porcine brain phosphatidylserine (PBPS, 99% purity) lipids were obtained from Avanti Polar Lipids (Alabaster, AL, USA), Inc. Buffer components, as well as D_2_O and ^15^N-Ammonium Chloride, were purchased from Merck and Sigma-Aldrich (Saint-Louis, MO, USA). D-^13^C-Glucose Monohydrate was obtained from Cambridge Isotope Laboratories (Tewksbury, MA, USA). NMR-tubes (5 mm, BMS-005TB) were obtained from Shigemi Inc (Allison Park, PA, USA). Tobacco Etch Virus (TEV) protease was expressed and purified according to [70]. Other materials for specific experiments are obtained as noted *vide infra*.

### 4.1. Primary Sequence Analysis

*In silico* analyses were performed using the A-Cage-C sequence “GSEQLTKAEVFRELKDLNLYIQWLKDGGPSSGRPPPSFLDDDLTDDIMAVKKILDKVG” as input. Three different metrics for protein disorder were assessed by DisEMBL (http://dis.embl.de/, accessed on 7 April 2018), using default settings. These metrics were COILS as defined by DSSP, and HOTLOOPS and REMARK465, which compare the query sequence with sequences in the Protein Databank with high B factors and missing coordinates in X-ray structures, respectively. These metrics are described in [39], and references therein. The Hydropathy Index was calculated using the approach of Tanford et al. [48], using a moving average over nine amino acids, and secondary structure was predicted using PSIPRED [38]. The online tool AMYLPRED2 automates the use of up to 11 different in silico methods to look for indications of amyloidogenic behaviour, and integrates the output as consensus segments in the query sequence [40]. AMYLPRED2 runs the individual methods using their default settings. This study used the following 10 out of 11 available methods: AGGRESCAN [71], AmyloidMutants [72], Amyloidogenic Pattern [73], Average Packing Density [74], b-strand contiguity [75], Hexapeptide Conformational Energy [76], NetCSSP [77], Pafig [78], SecStr [79], and TANGO [80,81]. Each method is described in their references, but brief documentation is also available at the AMYLPRED2 website, http://aias.biol.uoa.gr/AMYLPRED2 (accessed on 7 April 2018).

### 4.2. Protein Expression and Purification

Expression was performed in *E. coli* BL21 Star™ (DE3) (Invitrogen™) using M9 minimal medium supplemented with ^13^C-glucose, ^15^N-ammonium chloride and 100 ug/mL ampicillin. Cultures were grown at 37 °C until OD_595_ reached 0.6, and protein expression was induced by 1 mM Isopropyl β-D-1-thiogalactopyranoside, IPTG. Expression was maintained at 26 °C for 3 h, and then the cells were pelleted by centrifugation (10,000× *g*, 7 min). The pellet was resuspended in lysis buffer (50 mM Tris-HCl, pH 7.5, 150 mM NaCl, 20 mM imidazole, 0.2% Triton X-100 (*v*/*v*), supplemented with Complete EDTA free protease inhibitor cocktail, 1 tablet per 50 mL (Roche, Basel, Switzerland) and chicken egg white lysozyme (1 mg/mL). The resuspended pellet was kept on ice for 20 min and sonicated for 2 min with 15-s intervals. The cell lysate was centrifuged (50,000× *g*, 20 min) and purified using a Ni-NTA column (Qiagen, Hilden, Germany) following manufacturers recommendations. Eluted A-Cage-C was buffer exchanged using PD-10 Desalting columns (Millipore, Burlington, MA, USA) to TEV cleavage buffer and the His-tag was cleaved off with TEV protease overnight at 4 °C on a rotating wheel (TEV: polypeptide 1:30 molar ratio). To remove the affinity tag and TEV protease, a second Ni-NTA purification was performed, and the flow through was collected and concentrated. Finally, size exclusion chromatography was performed on a ÆKTA Purifier fitted with a Superdex 75 16/60 HiLoad prep grade (GE Healthcare, Chicago, IL, USA) equilibrated with 50 mM Tris-HCl, 300 mM NaCl, at pH 8.0.

### 4.3. Preparation and Characterization of Bicelles 

Bicelles of at a total lipid concentration 200 mM were prepared using deuterated and non-deuterated lipids for the lamellar and hexagonal part of the disk, D22-DHPC and D54-DMPC, and DHPS and DMPS. The bicelle *q*-value, a measure of the disk elongation [47], was set as ([D53-DMPC] + [DMPS])/([D22-DHPC] + [DHPS]) equal to 0.4. The correct amounts of D54-DMPC, DMPS, D22-DHPC and DHPS, were dissolved in chloroform and mixed in a glass tube. Chloroform was removed by rotavapor (glass tube in 37 °C water bath) operating at 100 mbar. After solvent removal, an appropriate volume of buffer (Citric Acid 8.9 mM, Trisodium Citrate 11.1 mM, NaCl 50 mM, NaZ 1 mM, D_2_O 10% (*v*/*v*), pH 4.5) was added. The glass tube containing the bicelles was placed in an ultrasonic bath for 20 min, resuspending the lipid film. The lipid mixture was then subjected to 10 cycles of freezing in liquid nitrogen, thawing in 60 °C water bath and vortexing at 1800 rpm for 40 s. The lipid mixture was further subjected to three cycles of 10 min incubation at 37 °C at 350 rpm and 10 min of incubation on ice. The final lipid concentrations of the bicelles corresponded to 130 mM D22-DHPC, 46 mM D54-DMPC, 12 mM DHPS and 12 mM DMPS.

Bicelles were characterized using Transmission Electron Microscopy (TEM). Samples were prepared on carbon mesh copper grids and stained with Uranyl Acetate (UA) (negative stain contrast for imaging). The TEM grid was placed in sample solution for 60 s and dipped in separate drops of water, then in 1% of UA for 60 s and then back in Milli-Q. Excess fluid was removed by touching a filter paper to the side of the grid. Prepared grids were left in a petri dish on a filter paper to dry. TEM imaging was performed on a Jeol Jem-1011, equipped with a tungsten filament, operating at an acceleration voltage of 80 kV. Images were captured by a MORADA camera, with optical image stabilization data system, and processed using iTEM. ImageJ was utilized for post-processing and analysis of the TEM images.

### 4.4. NMR Acquisition and Resonance Assignment

All NMR-experiments were acquired at 298 K and processed using TopSpin 2.1. The ^31^P-NMR experiments were carried out on an AV500 (Bruker) instrument, with a field strength of 11.7 T and fitted with a 5 mm Broadband Observe (BBO) probe. The carrier frequencies for the respective nuclei channels were 500.1 MHz and 202.2 MHz for ^1^H and ^31^P, respectively. Bicelle-containing samples were prepared by diluting a set amount of bicelles in NMR buffer at pH 4.5 (citric acid/citrate, 50 mM NaCl, 1 mM NaZ, 10% (*v*/*v*) D_2_O) to a volume of 500 µL, so that a final lipid concentration of 45 mM was achieved. The samples were transferred to a 528-PP-7 NMR tube and subjected to standard 1D proton-decoupled ^31^P-NMR experiments, using a pulse width of 13.3 µs, 64 scans, a relaxation delay of 4 s, and spectral width of 20 ppm. Inverse gated proton decoupling was achieved using a Waltz16 pulse-train with a 100 µs duration. The FID treated with a 1.2 Hz line broadening window function prior to Fourier transformation. Further information on the NMR acquisition parameters can be found in Appendix A.

^1^H-^15^N HSQC and triple-resonance NMR experiments were acquired on a Bruker Avance spectrometer operating at a proton frequency of 600.13 MHz (14.4 T) fitted with a cryogenically cooled 5 mm-TCI probe with pulse field gradients along the z-axis. The NMR sample used for resonance assignment contained 1.6 mM ^13^C-^15^N uniformly labelled A-Cage-C in citrate buffer at pH 4.5 or pH 7.0, containing 50 mM NaCl, 1 mM NaZ and 10% (*v*/*v*) D_2_O. To characterize the protein in the presence of bicelles, ^1^H-^15^N HSQC spectra of 1.1 mM A-Cage-C in the presence of DMPC: DMPS (*q =* 0.4) bicelles at a final lipid concentration of 100 mM were acquired. The following spectra were acquired for the purpose of resonance backbone assignment: HNCA, HNCO, CBCA(CO)NH, CBCANH, Sequence-specific backbone resonance assignment was performed using CARA (Computer Aided Resonance Assignment) version 1.8.4.2 [82]. Based on Cα and Cβ chemical shifts, the secondary structure propensity score per A-Cage-C residue was calculated using the Secondary Structure Propensity (SSP) program [83]. For more NMR acquisition parameters, refer to Appendix A. Assignments relevant for the work is submitted to the BMRB with accession code 26892.

### 4.5. Flexible-Meccano and Cluster Analysis

Ensembles of 10,000 A-Cage-C structural conformers were calculated in the Flexible-Meccano program using the amino acid sequence and SSP scores as input parameters [56]. In the case of the bicelle-associated data at pH 4.5, dihedral angles from a TALOS analysis were substituted for the SSP analysis. The 10,000-conformation ensembles of A-Cage-C derived in this way were converted to topology and coordinate files using the Amber LEaP program [84]. This was done using data acquired in the presence and absence of bicelles at pH 4.5, one 10,000-conformation ensemble for each condition. Average linkage hierarchical agglomerative clustering with distances based on peptide backbone best-fit RMSd for cluster counts ranging from 2 to 13 was performed with Amber Cpptraj [85]. The clustering results were subsequently assessed via the Davies–Bouldin index (DBI) [57,58] and pseudo F-statistic (pSF) [58] metrics to determine the most reasonable cluster count for each ensemble of Flexible Meccano conformations. Members of each cluster were RMSD fitted to the backbone (C, N, Cα) of the relevant cluster centroid and visualized in Visual Molecular Dynamics (VMD) [86], and The PyMOL Molecular Graphics System, Version 1.2.2, Schrödinger, LLC. Identified clusters were further characterized by estimation of secondary structure content using the DSSP method [87], average surface area with the LCPO method [88] and average radius of gyration calculated by Cpptraj [85].

### 4.6. Molecular Dynamics Simulations

Two conformations, the ones with the longest and shortest centre-of-mass distance from the first to the last residue, were selected from the Flexible Meccano ensemble of A-Cage-C and subjected to MD simulations in solution. In each case, the peptide was assigned ff14SB parameters [89] and placed in a truncated octahedral TIP3P [90] water box containing K^+^ counterions to obtain a neutral net charge. Each system was subjected to the following regime: (i) 5000-step minimization (steepest descent and conjugate gradient, for the first and last half of the minimization, respectively); (ii) 5 ps heating from 0 to 100 K at constant volume; (iii) 100 ps heating from 100 K to 310 K at constant pressure; (iv) unrestrained simulation for about 500 ns at 310 K and constant pressure. Ten kcal mol^−1^Å^−2^ restraints were applied to the peptide backbone during both heating steps.

To find a reasonable starting conformation for simulation in the presence of DOPC and DOPC:DOPS lipid bilayers, we applied the following strategy. The two simulations of A-Cage-C in solution were combined and subjected to the same clustering approach as described above for Flexible-Meccano output, from which a representative frame from the largest cluster was extracted (by lowest cumulative distance to all other frames in the cluster). The resulting A-Cage-C conformation was placed in the aqueous phase of two bilayer systems containing: (A) 232 DOPC lipids [91] and (B) a 4:1 DOPC:DOPS bilayer [91,92] composed of 240 lipids. Both systems were solvated by about 28,000 TIP3P water molecules [90] and contained 150 mM NaCl [93] as well as additional Na^+^ counterions for neutralization of the net charge. The simulated volume was large enough to allow full peptide unfolding with a minimum of 20 Å between any solute atom and the edge of the octahedral water box. The same minimization/simulation regime as described above was performed, with two notable differences: semi-isotropic pressure coupling was used in step (iii) and (iv) while isotropic pressure was applied for A-Cage-C in solution, and each of the bilayer-containing systems were simulated for 800 ns in step (iv).

All simulations were run with the GPU-accelerated version of AMBER18 [94,95,96,97] under periodic boundary conditions. Simulation including lipids used the lipid force field Lipid14 [89,90]. A 10 Å nonbonded cut-off truncated the van der Waals interactions, while electrostatic energies were evaluated by the particle mesh Ewald (PME) method [98]. Application of SHAKE [99] for constraining bond lengths involving hydrogen enabled the use of 2 fs time steps. Temperature was regulated by the Langevin thermostat [100] with a 1.0 ps^−1^ collision frequency, and pressure was maintained at a reference value of 1 bar and a relaxation time of 1.0 ps by the Berendsen barostat [101].

## Figures and Tables

**Figure 1 molecules-26-03607-f001:**
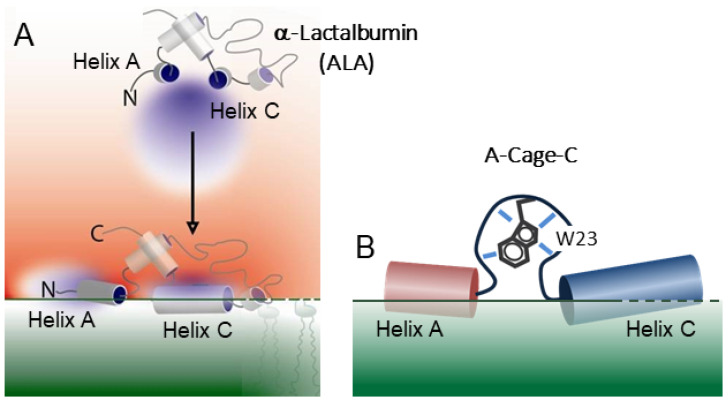
α-Lactalbumin and A-Cage-C. (**A**) Depiction of full-length ALA and the phospholipid membrane (green shading). The proton-gradient (red colour gradient) drives the interaction between the ALA and acidic membranes near the lipid bilayer at mildly acidic pH conditions (pH 4.5) by successive protonation at key residues in the A and C helix of the full-length protein [18]. Peptides derived from these helices can also form pore-like lesions when co-deposited as parts of lipid monolayers [33]. (**B**) Cartoon sketch of the A-Cage-C, derived from full-length ALA. Joining the two helices together with a short peptide to allow them to act together in the absence of the tertiary structure of ALA [34].

**Figure 2 molecules-26-03607-f002:**
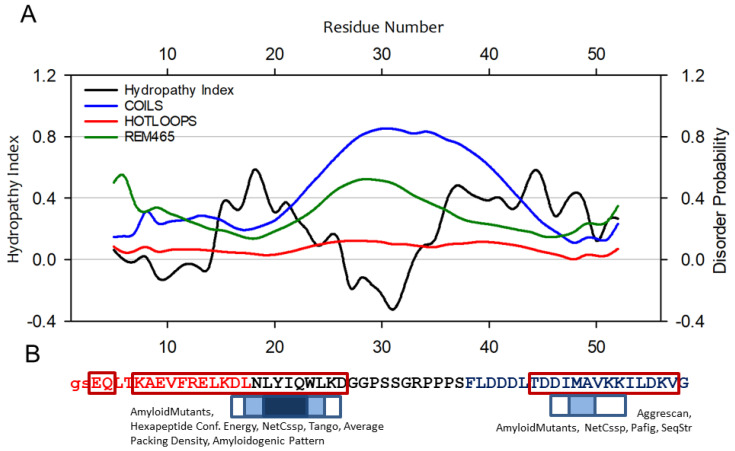
A-Cage-C and some of its predicted properties. (**A**) The Hydropathy Index and disorder probability of A-Cage-C. Hydropathy Index was calculated using the approach of Tanford et al. [37], and plotted over a moving average of 9 amino acids. Disorder was predicted using the DisEMBL server [39], which provide three outputs COILS; HOTLOOPS and REMARK465. (**B**) The sequences highlighted in red and blue corresponds to the membrane binding segments Helix A and C of BLA, respectively. The sequences in black correspond to the spacer segment based on the Trp-Cage fold. The amino acids within dark red boxes are predicted to be helical using PSIPRED [38]. The two lower-case amino acids at the N-terminal are left after TEV cleavage during purification. The sequence is also annotated with in silico-prediction of amyloid-forming tendencies using AMYLPRED2. The results shown are based on the consensus of 10 methods that AMYLPRED2 runs and compares, which are listed and referenced in the Materials and Methods section. The boxes above the sequence indicate consensus between at least three methods (box with white fill), at least five methods (box with light blue fill), and consensus in all 10 methods (dark blue fill). The text above each box indicates which individual approaches are involved in giving a consensus based on at least five of the methods indicated.

**Figure 3 molecules-26-03607-f003:**
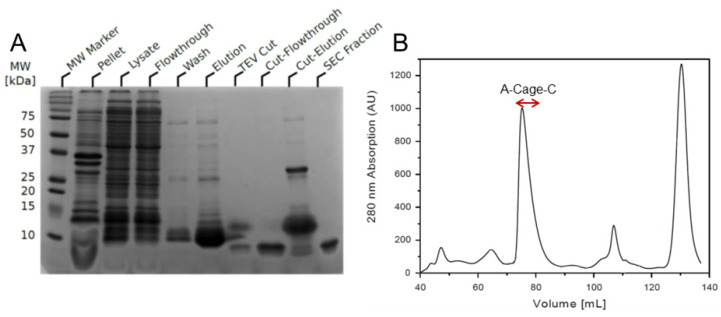
Expression and purification of ^13^C, ^15^N isotopically labelled A-Cage-C. (**A**) SDS-PAGE of A-Cage-C. SDS-PAGE of samples collected during purification of A-Cage-C. (**B**) Size-exclusion elution profile using a Superdex 75 16/60 HiLoad prepgrade. Volume fractions containing peptide were between 74–82 mL, indicated by the red arrow brackets.

**Figure 4 molecules-26-03607-f004:**
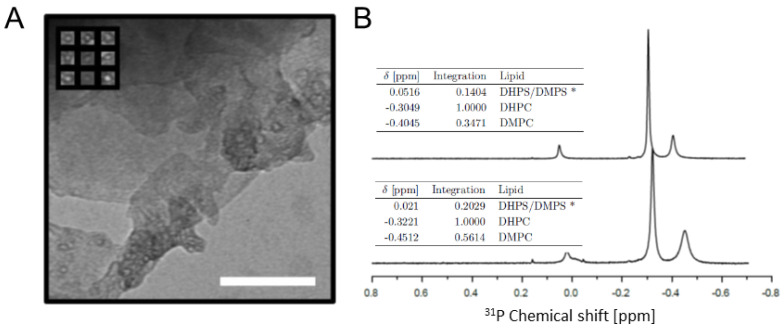
Characterization of bicelles: (**A**) TEM image of bicelles adsorbed to carbon mesh copper grid and stained with UA. The inset contains a selection of bicelle images. The scale bar is 100 nm. (**B**) ^31^P-NMR of bicelles and A-Cage-C in the presence of bicelles. The plot shows proton decoupled ^31^P-NMR spectra of the following samples. Top: 45 mM bicelles, pH 5.5. Bottom: 100 mM bicelles and 1.1 mM A-Cage-C, pH 4.5. Table insets list DHPS, DMPS and DHPC chemical shifts for each major peak in the spectra, as well as peak integration volumes relative to the DHPC peak volume.

**Figure 5 molecules-26-03607-f005:**
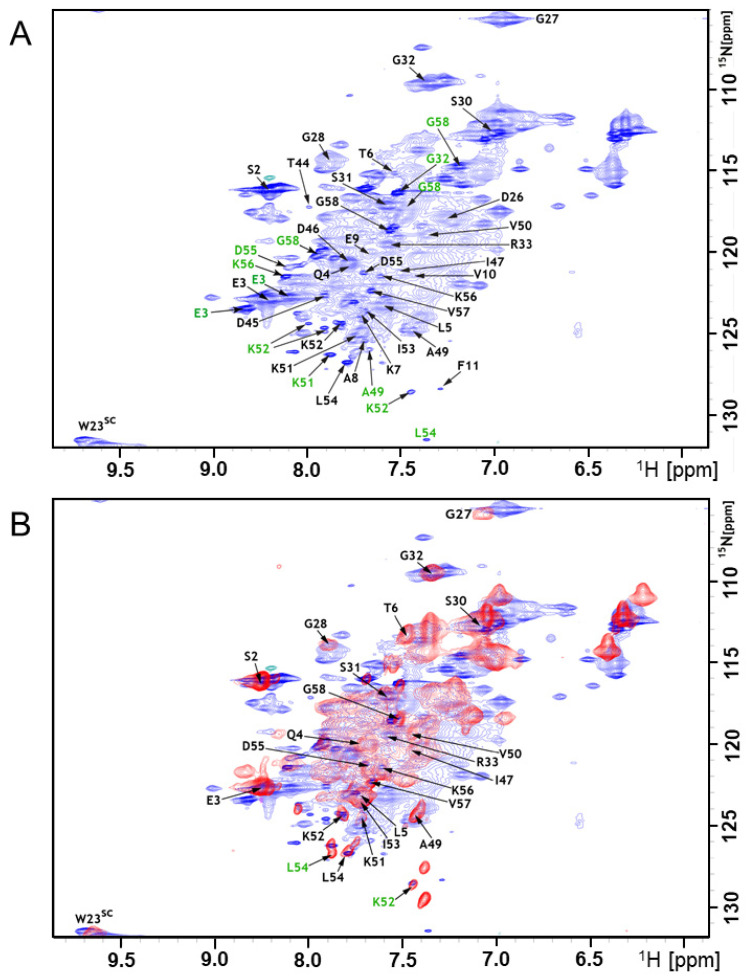
Backbone resonance assignment at pH 4.5 (**A**) ^1^H-^15^N HSQC in the absence of bicelles with assignments indicated by residue number and arrows in black. Residues E3, G32, A49, K51, K52, L54, D55, K56, G58, highlighted in green, showed additional signals in the presence of bicelles. (**B**) Overlay of the ^1^H-^15^N HSQC in the absence (blue) and presence of bicelles (red) with assignments indicated by residue number and arrows in black. Residues K52 and L54, highlighted in green, showed minor additional signals as indicated.

**Figure 6 molecules-26-03607-f006:**
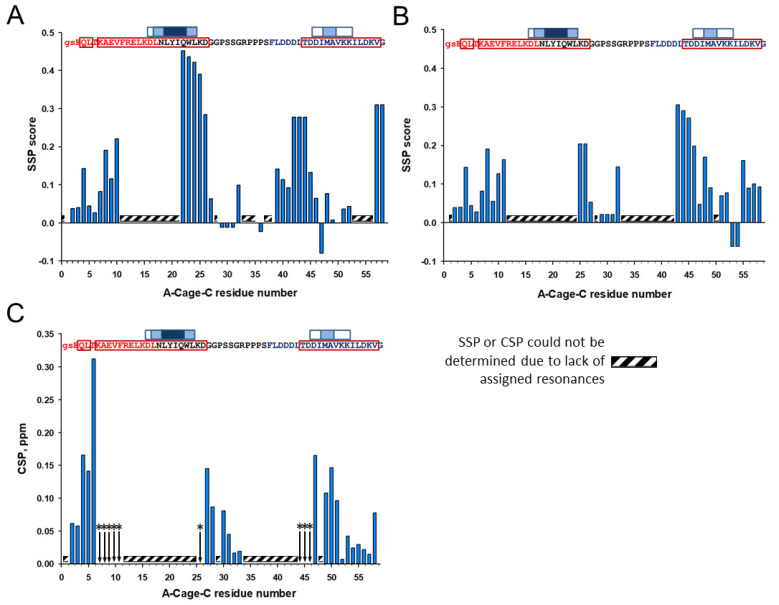
Effects of pH changes and addition of bicelles. (**A**) Secondary structure propensity (SSP) score calculated based on the assigned residues of A-Cage-C at pH 7.0 in the absence of bicelles. (**B**) SSP score calculated based on the assigned residues of A-Cage-C at pH 4.5 in the absence of bicelles. (**C**) Chemical shift perturbation (CSP) analysis obtained from comparison of the A-Cage-C peptide resonance assignment in the absence and presence of bicelles at binding conditions (pH 4.5). Amino acids at positions indicated by the arrows with stars (*) showed signal broadening beyond detection upon binding to bicelles. Panel insets: A-Cage-C Sequence information aligned with horizontal axis residue number. Red, black, and blue capital letters indicate Helix A, Trp-Cage, and Helix C motifs; lower case letters indicate two remaining residues from TEV cleavage; Red frames indicate predicted helices; White, light blue and dark blue boxes indicate increasing AMYLPRED2 consensus. For further details, see Figure 2 and Materials and Methods.

**Figure 7 molecules-26-03607-f007:**
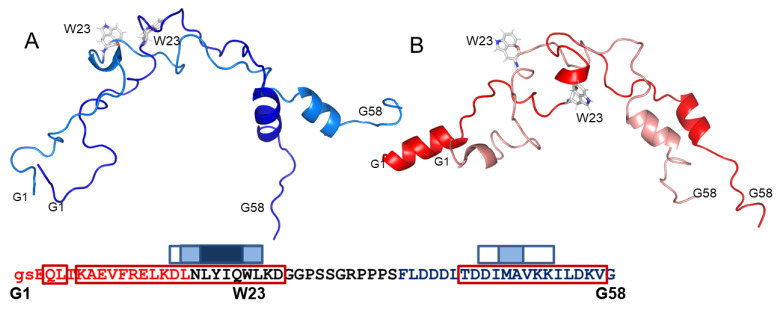
Centroid representations of two A-Cage-C clusters in the absence (blue shades) and presence (red shades) of bicelles at pH 4.5. (**A**) Centroid representation of Cluster 1 (Dark Blue; 6349 structures) and Cluster 2 (Light Blue, 3651 structures) of the Flexible-Meccano analysis. The input of NMR data accumulated at pH 4.5 in the absence of bicelles were used. (**B**) Centroid representation of Cluster 1 (Dark Red, 5105 structures) and Cluster 2 (Light Red, 4891 structures) of the Flexible-Meccano analysis. The input of NMR data accumulated at pH 4.5 in the presence of bicelles were used. For both panels, all structures were loaded in PyMOL, aligned to Cluster 1 of Panel A, and represented as backbone cartoons. The termini residues, G1 and G58, as well as and the Cage Tryptophan, W23, are indicated. Bottom inset: A-Cage-C Sequence information aligned with horizontal axis residue number. Red, black, and blue capital letters indicate Helix A, Trp-Cage, and Helix C motifs; lower case letters indicate two remaining residues from TEV cleavage; Red frames indicate predicted helices; White, light blue and dark blue boxes indicate increasing AMYLPRED2 consensus. For further details, see Figure 2 and Materials and Methods.

**Figure 8 molecules-26-03607-f008:**
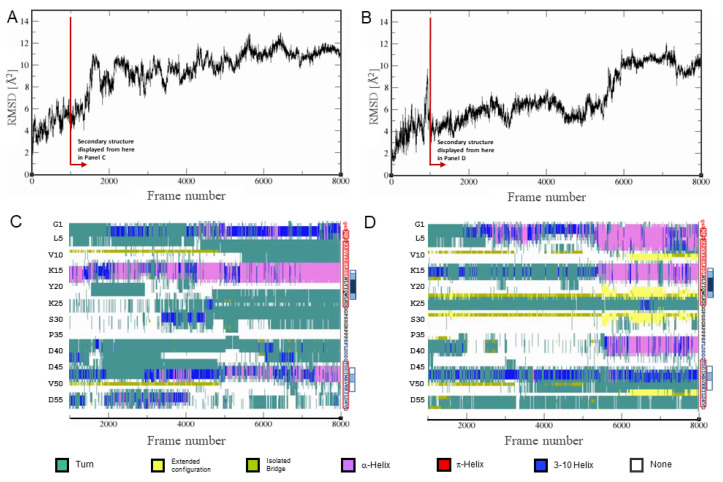
MD simulation of A-Cage-C in water and the presence of DOPC and DOPC:DOPS bilayers. (**A**) Simulation of A-Cage-C in water and the presence of DOPC bilayers. The Cα RMSD values are calculated relative to the starting structure, selected as described in the Materials and Methods section. Each frame of simulation represents 100 ps, and the total simulation time was 800 ns. The red arrow indicates from which frame secondary structure is presented in the panel below. (**B**) Simulation of A-Cage-C in water and in the presence of DOPC:DOPS (4:1) bilayers. The Cα RMSD values are calculated relative to the starting structure, selected as described in the Materials and Methods section. Each frame of simulation represents a 100 ps, and the total simulation time was 800 ns. The red arrow indicates from which frame secondary structure is presented in the panel below. (**C**) A-Cage-C simulated in the presence of DOPC bilayers, where A-Cage-C secondary structure determined by DSSP as a function of simulation time and residue position. (**D**) A-Cage-C simulated in the presence of DOPC bilayers, where A-Cage-C secondary structure determined by DSSP as a function of simulation time and residue position. For the two bottom panels, sequence information aligned with the vertical axis can be found to the right at each panel. Red, black, and blue capital letters indicate Helix A, Trp-Cage, and Helix C motifs; lower case letters indicate two remaining residues from TEV cleavage; Red frames indicate predicted helices; White, light blue and dark blue boxes indicate increasing AMYLPRED2 consensus. For further details, see Figure 2 and Materials and Methods.

## Data Availability

Assignments relevant for the work is submitted to the BMRB with accession code 26892.

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
