# Peer review of "Investigating the Disordered and Membrane-Active Peptide A-Cage-C Using Conformational Ensembles"

_molecules, 2021, doi:10.3390/molecules26123607_

Round 1
Reviewer 1 Report
As the authors admit, their initial aim was to study membrane-induced oligomerization of the A-Cage-C polypeptide. After finding that the polypeptide does not polymerize, the authors decided to investigate general behavior of the polypeptide. The paper thus not answer the original question. Nevertheless, general conformational analysis may provide information that contributes to our understanding of behavior of membrane-interacting peptides. Corrections suggested by the reviewer and his comments are listed below.
(1) l. 152 The sentence "Bicelles are.." suggests that all bicelles are isotropically tumbling objects, which is not true. This should be clarified and small isotropic bicelles should be distinguish from frequently used large, anisotropic bicelles (with high q values).
(2) l. 156 vs. 512 The abbreviation TEM is explained as "Tunnelling Electron microscopy" (= Scanning tunneling microscopy?), whereas (probably correct) "Transmission electron microscopy" is mentioned in Materials and Methods. This should be clarified/corrected.
(3) l. 278 Fig. S4 suggests that the N-terminal and C-terminal (red and blue) regions of Cluster 2 directly interact. Is it true? If so, is there any experimental evidence for such interactions?
(4) l. 573 It is well-known that the TIP3P water model significantly underestimates London dispersion forces and leads to formation of artificial compact structures in MD simulations of intrinsically disordered proteins (e.g., Piana et al., J. Phys. Chem. B. 119 (2015) 5113–5123, Zapletal et al., Biophys. J. 118 (2020) 1621–1633). How is the use of TIP3P justified in the presented study? Could its use contribute e.g. to the differences presented in Fig. S5? Discussion of this issue should be included.
Reviewer 2 Report
I hope the authors can address the following points:
- Line 107
There is no Figure 1C.
- Line 117 "The amino acids within dark red boxes are predicted to be helical using PSIPRED"
This is found in Figure 2B, not in 2A as indicated.
- Line 592
Which AMBER version was used?
- What lipid force field was used?
- Line 573
How large was the water box/How far was the peptide from the edge of the water box?
- Line 574
What are the positive counterions specifically?
- Line 575
What energy minimisation algorithm was used?
- A-Cage-C is known to interact with membranes when the pH is low, but MD simulations were done at neutral pH. It is not surprising that the authors did not observe adhesion of the protein with the membrane. Hence, there is no need to speculate on the inability of bicelles to interact with A-Cage-C at the end of the discussion. The authors could instead try to determine from the simulations which parts of A-Cage-C or residues are preventing its association with the membrane.
Round 2
Reviewer 2 Report
The authors have satisfactorily addressed my comments.